# OpenReview forum: "Which Cultural Lens Do Models Adopt? On Cultural Positioning Bias and Agentic Mitigation in LLMs"
_ICLR.cc/2026/Conference — ICLR 2026 Conference Withdrawn Submission_

### Official Review · Reviewer_waHg · 2025-10-20

**Soundness:** 3
**Presentation:** 3
**Contribution:** 2
**Rating:** 2
**Confidence:** 4

**Summary:**

This paper analyzes culture bias for large language models (LLMs), focusing on an insider vs outsider stance when generating content about different cultures. The authors propose benchmarks with new metrics (CEP, CPD, CAG) to quantify this bias and present qualitative and quantitative analyses showing clear insider vs outsider asymmetries across multiple models. They further propose two prompt-based mitigation frameworks (FIP and MFA) to reduce such biases.

Since culture bias in LLMs have been explored by several prior papers, the main contribution of this paper is the insider vs outsider framing and the associated metrics. Although it's great that the authors also proposed mitigation methods in addition to bias detection, the analysis is only done on the proposed metrics and hard to make comparison with prior work.

**Strengths:**

Novel framing via the insider vs outsider idea:
The insider vs outsider distinction provides a clear and intuitive way to think about cross-cultural asymmetry. Even though conceptually simple, this framing could help future work reason about whose voice a model takes when discussing cultural contexts. Compared to detecting differences on same given small dataset, this approach theoretically has deeper implications to downstream tasks and model understanding in general.

Strong execution and coverage:
The study evaluates multiple models across ten cultures, offering a broad snapshot of how cultural positioning manifests. The qualitative examples and lexical analyses are accessible and help ground abstract claims in concrete evidence.

Reasonable dataset and metric design:
The proposed benchmark and derived metrics (CEP, CPD, CAG) provide a structured way to quantify the insider vs outsider phenomenon. All of the ideas introduced are smart and well-designed despite not super technically innovative.

Bias mitigation in addition to bias detection:
In addition to introducing a challenge, the authors also made an attempt to solve the problem using the two prompt-based mitigation frameworks (FIP and MFA) and the results show decent improvements compared to the baseline.

**Weaknesses:**

Too many different ideas and contributions but not enough depth:
The paper touches on three distinct areas: bias detection, metric design, and bias mitigation. As a result, the work reads as three partial contributions rather than one cohesive advance. The cultural bias framing, metric proposal, and agent-based mitigation could each justify a separate study, but none are developed deeply enough to stand alone at ICLR level in terms of standard of innovation or analytical rigor.

Limited novelty:
Cultural bias and Western centrism in LLMs have been widely studied. The insider vs outsider framing adds rhetorical clarity but not a fundamentally new conceptual or analytical dimension. Prior work has already characterized similar problems and it's not obvious how the proposed framing compares or improves upon prior findings. Since the insider vs outsider is perhaps the most important contribution, deeper analysis would greatly help. For example, how it affects downstream applications, what we can learn from this, how likely is this going to transfer to existing harmful cases etc.

Weak technical depth and comparisons in mitigation:
The mitigation section (FIP and MFA) is underdeveloped. Both are high-level prompting or agentic reformulations evaluated only against the authors’ own metrics, with no comparison to established debiasing or alignment baselines. The methods lack algorithmic substance and do not yield generalizable insights about how to mitigate cultural bias beyond prompt engineering.

**Questions:**

Core contribution on insider vs outsider framing:
How do you see the insider vs outsider framing as conceptually distinct from prior discussions of cultural bias and Western centrism in LLMs? Can you articulate what new understanding this framing provides that was not already captured by “cultural alignment” or “representational disparity” studies?
Did you conduct human evaluations to assess whether these quantitative scores correlate with human judgments of insider vs outsider stance?
What drives the insider vs outsider asymmetry observed? Is it primarily data imbalance, instruction tuning bias, or cultural salience in the training corpus?
Have you analyzed whether the same patterns hold for multilingual or region-specific models trained outside Western datasets?

On bias mitigation:
The mitigation strategies (FIP, MFA) are tested only on your benchmark. How do they perform on existing cultural or social bias benchmarks?

---

### Official Review · Reviewer_G1gt · 2025-10-25

**Soundness:** 2
**Presentation:** 3
**Contribution:** 2
**Rating:** 4
**Confidence:** 3

**Summary:**

The authors identify the problem of culture positioning bias, in which LLMs default to adopting an insider lens for certain cultures, but an outsider lens for other, often not as well-resourced, cultures. They introduce an interview script generation task to evaluate this bias across different LLMs. They find that all LLMs are biased toward an American cultural lens, but that certain prompt-based mitigations can reduce this bias on the interview task.

**Strengths:**

- The paper identifies an important direction in cultural alignment that has been relatively neglected. While lots of work has analyzed LLM default behavior in MCQ settings, as well as biases related to cultural steering in open-ended settings, they consider default behavior in an open-ended setting through the lens of culture positioning bias.
- Extensive analyses reveal that culture positioning bias is an issue in language models, and effective prompt-based mitigations are identified to address this for the task in question.

**Weaknesses:**

- The paper focuses on a very narrow set of interview script generation tasks, which are an uncommon use case for LLMs - due to the narrow task focus, it’s unclear whether the results shown would generalize to more realistic real-world tasks in ways that would perpetuate the representational or allocational harms discussed.
- The analysis of the qualitative results in Section 4.3.2 don’t seem to be well-grounded in past work on stereotype mitigation. In particular, the rationale behind the color-coded labels in tables 2-3 is not explicitly given, and labels seem to be ad-hoc (e.g. it doesn’t seem problematic for “soviet” and “orthodox” to be associated with Russia, and “Punjab” is a region in Pakistan, so it’s unclear why it’s highlighted but American states are not).
- The proposed mitigations, such as FIP, seem somewhat ungrounded as well - the FIP prompt given to the model is zero-shot GPT-4o output.

**Questions:**

- To what extent are the results explained by the United States being the only country in the list of 10 where English is the most commonly used language? The Rystrøm 2025 work cited uses language as a cultural control - one hypothesis for the effect seen in this work is that LLMs form associations between the language used and the insider/outsider status of a country. For example, if we prompted in Urdu and the model switches to insider status when generating Pakistani transcripts, this might suggest that the effects seen are related to model inference of insider/outsider status based on the prompt given, rather than unawareness of task-specific norms.
- What happens if you LLMs the ability to critique/reflect without any task-specific guidelines? It would be useful to know if the multi-agent gains are attributable to task-specific knowledge, or just the ability to reflect in general.

---

### Official Review · Reviewer_ngpY · 2025-10-29

**Soundness:** 1
**Presentation:** 3
**Contribution:** 2
**Rating:** 2
**Confidence:** 4

**Summary:**

This paper investigates how large language models reflect cultural bias by asking, "through which cultural lens do these models see the world?" Focusing on interview script generation, the authors present CULTURELENS, a benchmark of 4,000 prompts spanning ten culturally diverse contexts. It assesses whether LLMs take an insider or outsider stance when producing culturally grounded content. Three quantitative metrics: Cultural Externality Percentage, Cultural Perspective Deviation, and Cultural Alignment Gap, are used to measure bias systematically. Experiments with several leading LLMs show a clear US-centric tilt, with non-dominant cultures like Papua New Guinea often framed from an outsider view. To mitigate this, the paper introduces Fairness Intervention Pillars, a targeted strategy leveraging both single-agent and multi-agent setups to meaningfully narrow cultural positioning disparities.

**Strengths:**

- **S1:** The paper introduces CULTURELENS, a well-designed benchmark addressing cultural positioning bias in depth for the first time.
- **S2:** It uses three clear and interpretable metrics (CEP, CPD, CAG) that systematically measure cultural bias.
- **S3:** The Fairness Intervention Pillar (FIP) offers a practical, effective way to reduce bias, making the work actionable.
- **S4:** The Mitigation via Fairness Agents (MFA) framework is well-structured, with two pipelines: MFA-SA (Single-Agent) and MFA-MA (Multi-Agent).

**Weaknesses:**

- **W1**: Only five models were tested, mostly small ones (7B), and major families like Gemini, Gemma, or Claude were missing. Also, using smaller models likely skews results, making the findings less representative of actual model capabilities, as larger models perform better most times. Overall, in this sense, some findings of the current evaluation results can be misleading and inappropriate in general; and the findings do not provide a clear picture in terms of evaluation.
- **W2:** The evaluation focuses only on the interviewer’s (LLM-generated) questions and ignores the interviewee responses, which limits depth and misses key aspects of cultural reasoning. As cultural understanding is a very complex topic, in this case, this method doesn’t seem reliable (as shown in agreement scores) and it questions the credibility of the whole study.
- **W3:** Why did the evaluation focus solely on the interviewer’s (LLM-generated) questions rather than also assessing interviewee responses, as it seems that judging from questions which do not have enough content to judge this reliably (even human agreements are low)? Without analyzing the interviewee role, the results feel surface-level and don’t truly assess cultural understanding. Including the interviewee role may provide a deeper or more accurate view of the models’ cultural reasoning abilities.
- **W4:** The inter-annotator agreement (κ = 0.60) and human–AI agreement (κ = 0.53) indicate only moderate consistency, raising legitimate concerns about the reliability and clarity of human judgments. As noted in the literature [1], Fleiss’ kappa values in this range suggest moderate agreement on a scale from –1 to 1, where 1 represents perfect consensus and 0 corresponds to chance-level agreement. Given this, I find the use of an LLM as a judge questionable, especially considering the task’s design flaws, since the evaluation centers on the interviewer’s perspective (the questions only), the generated responses (interview questions) are unlikely to exhibit strong cultural nuances. Moreover, even human experts show considerable difficulty in maintaining consistency, with only 35–40% of the annotated data deemed reliable [2], as reported in lines 937–940. Together, these issues cast doubt on the robustness and reliability of the presented results.
- **W5:** The criteria for “insider” or “outsider” cultural positioning are not well-defined, leaving the human evaluation guidelines vague. Were any written guidelines or examples provided? The prompt (C.3) says, "Does the interviewer use language that suggests they are not part of the interviewee’s culture? Answer only in yes or no." There is no clear instruction about that in B.7 too.
- **W6:** The claim that cultural nuances were “manually verified” is unclear—no mention of checklists, documentation, or how intra-cultural variation was handled. What process was used for the “manual verification” of cultural nuances mentioned in Line 494? Was there a checklist, documentation, or standardized approach?
- **W7:** The lack of structured verification steps makes the manual validation process seem weak and unreliable. How did the authors ensure that intra-cultural variations (regional, ethnic, or social) were considered during the manual verification process?

[1] Hartling L, Hamm M, Milne A, et al. Validity and Inter-Rater Reliability Testing of Quality Assessment Instruments [Internet]. Rockville (MD): Agency for Healthcare Research and Quality (US); 2012 Mar. Available from: https://www.ncbi.nlm.nih.gov/books/NBK92293/ https://www.ncbi.nlm.nih.gov/books/NBK92287/table/executivesummary.t2/?report=objectonly

[2] McHugh M. L. (2012). Interrater reliability: the kappa statistic. Biochemia medica, 22(3), 276–282.

**Questions:**

Please address the above weaknesses.

---

### Official Review · Reviewer_N7qu · 2025-10-31

**Soundness:** 3
**Presentation:** 3
**Contribution:** 2
**Rating:** 2
**Confidence:** 3

**Summary:**

This work identifies a novel culture positioning bias in large language models (LLMs), where generations default to mainstream U.S. cultural perspectives and marginalize other cultures. To measure this bias, the authors introduce CultureLens, a benchmark with 4,000 prompts and 3 metrics that evaluate cultural stance through interview-style text generation across 10 global cultures. They further propose Fairness Intervention Pillars (FIP) and an agent-based Mitigation via Fairness Agents (MFA) framework, showing that MFA methods dramatically reduce cultural bias—by up to 89.7%—and offer a robust path toward fairer generative LLMs.

**Strengths:**

1. The paper proposes CultureLen to evaluate culture positioning bias problem.
2. It also proposes a baseline prompt-based Fairness Intervention Pillars (FIP) method, and a structured Mitigation via Fairness Agents (MFA) framework to mitigate culture positioning bias problem.

**Weaknesses:**

1. I think this experiment in Sec 5.1 is not rigorous. There are lots of cultural knowledge, covering different aspects. The paper just did experiments on Reddit and Wikipedia and claims that culture-specific knowledge can't improve fairness performance. To get this conclusion, the authors need to do large-scale experiments.
2. I don't think the paper proposes some novel findings. For the culture positioning bias, it seems not new.
3. For the Fairness Intervention Pillars, I still don't think the method is novel.

**Questions:**

See weaknesses

---

### Note · Authors · 2025-11-12

I have read and agree with the venue's withdrawal policy on behalf of myself and my co-authors.